

# Diversity of bacteria associated with lichens in Mt. Yunmeng in Beijing, China

Yimeng Li[1], Yinzhi Huang[2], Torsten Wronski[3] and Manrong Huang[1]

[1] Department of Life Sciences, National Natural History Museum of China, Beijing, China
[2] The Experimental High School Attached to Beijing Normal University, Beijing, China
[3] School of Biological and Environmental Sciences, Liverpool John Moores University, Liverpool, United Kingdom

## ABSTRACT

Lichens host highly complex and diverse microbial communities, which may perform essential functions in these symbiotic micro-ecosystems. In this research, sequencing of 16S rRNA was used to investigate the bacterial communities associated with lichens of two growth forms (foliose and crustose). Results showed that Pseudomonadota, Actinomycetota and Acidobacteriota were dominant phyla in both types of lichens, while Acetobacterales and Hyphomicrobiales were the dominant orders. Alpha diversity index showed that the richness of bacteria hosted by foliose lichens was significantly higher than that hosted by crustose ones. Principal co-ordinates analysis showed a significant difference between beta diversity of the foliose lichen-associated bacterial communities and those of crustose lichen-associated ones. Gene function prediction showed most functions, annotated by the lichen-associated bacteria, to be related to metabolism, suggesting that related bacteria may provide nutrients to their hosts. Generally, our results propose that microbial communities play important roles in fixing nitrogen, providing nutrients, and controlling harmful microorganisms, and are therefore an integral and indispensable part of lichens.

## INTRODUCTION

Lichens, a symbiotic complex that consists of two partner organisms, *i.e.*, a fungal partner (mycobiont) and a photoautotrophic partner (a green alga or cyanobacterium, photobiont), are considered among the most endurable organisms on earth (*Nash, 2008*). They can thrive in most terrestrial environments, including the Antarctic, the Arctic, deserts, *etc.*, where environmental conditions are too harsh to be tolerable for most plants (*Øvstedal & Smith, 2001*; *Printzen, 2008*). In particular, lichens that dwell on rocks accelerate the weathering process of their substrates and facilitate pedogenesis through mechanical and chemical processes (*Gayathri & Swamy, 2012*; *Swamy, Gayathri & Devaraja, 2016*), therefore creating suitable environment for other organisms, such as mosses, microfauna, or other microbes (*Grube & Berg, 2009*; *Grube et al., 2009*; *Bates et al., 2011*).

Microbial communities hosted in lichen thalli were investigated by various authors, using cultural approaches (*Grube et al., 2009*), molecular fingerprinting methods

Corresponding author
Manrong Huang,
hmanrong@126.com

(*González et al., 2005*), fluorescence *in situ* hybridization staining (*Erlacher et al., 2015*), and, more recently, next-generation sequencing approaches (*Lee et al., 2014*; *Hodkinson et al., 2012*; *Swamy & Gayathri, 2021*). *Grube et al. (2015)* concluded that lichen-associated microbial communities may play important roles in the health, growth, and fitness of their hosts (*Bosch & McFall-Ngai, 2011*). Consequently, a growing number of researchers urges to redefine lichens as holobionts that contain a dominant fungus, one or more photosynthetic partners, and a plethora of other microorganisms (*Allen & Lendemer, 2022*; *Hawksworth & Grube, 2020*; *Simon et al., 2019*).

Factors that affect the composition of lichen-associated microbial communities are not yet fully understood and require further investigations. *Hodkinson et al. (2012)* suggested that the composition of bacterial communities depends largely on the availability of fixed carbon and nitrogen, as well as on the selective pressure induced by mycobionts through the production of secondary metabolites with antimicrobial activity. Other researchers proposed that either host species, growth form, substrate and geography might be putative factors shaping the microbial community (*Park et al., 2016*; *Fernández-Brime et al., 2019*; *Alonso-Garcia & Villarreal, 2022*).

When we were mapping the lichens biodiversity of Beijing, a heavily urbanized ecosystem with a degraded lichens diversity, we also collected materials from Mt. Yunmeng, an area that was gradually restored in recent years (http://www.yunmengshan. org.cn/ymsjqjs). Here, we glimpsed the rare opportunity to evaluate the diversity of lichen-associated bacterial communities in an ecosystem during its restoration process, and therefore enhance our understanding of the maintenance and dynamics of lichen-associated microbial communities and the relationship with their hosts. Thus, we conducted DNA extraction, 16S rRNA gene amplification and sequencing on the material gathered, with the objective to compare the composition of the associated microbial communities between two growth forms of lichens, *i.e.*, foliose lichens and crustose lichens.

## MATERIALS AND METHODS

### Study area

Lichen samples were collected at Mt. Yunmeng (40°26′N–40°38′N, 116°30′E–116°50′E), which is located in the northwest of Miyun District, Beijing. The climate of this area is characterized by warm temperate semi-humid continental monsoon. The average annual temperature is 10.9 °C, the average annual precipitation ca. 700 mm, with 76% of the precipitation occurring from June to August. The vegetation was severely destroyed by human activities in the past, but the government classified Mt. Yunmeng and its adjacent areas as a nature reserve and a national forest park in the early 1990s. Since then, a secondary temperate deciduous broadleaf forest developed and became dominant, with planted coniferous forest interspersing some areas. Today about 90% of the Mt. Yunmeng area is covered by forest (http://www.yunmengshan.org.cn/ymsjqjs).

While most parts of Mt. Yunmeng were classified as nature reserve, its eastern part has developed into a series of scenic spots which attract large numbers of tourists. One of these spots is the Black Dragon Pool (160 to 380 m a.s.l), which is a 4.5 km long canyon, with a

small creek and a riverine forest on both sides. Tourists' activities may have destroyed or negatively affected the local lichen communities.

## Collection of materials

Lichen samples were collected in October starting from the Black Dragon Pool, crossing the border between the scenic spot and the nature reserve, and ending at the heart of nature reserve at about 1,130 m a.s.l. Lichen samples were identified referring to *Brodo, Sharnoff & Sharnoff (2001)* and *Nash et al. (2007)*. In total, twenty samples were obtained with three from the scenic spot (F1–F3) and seventeen from the nature reserve (F4–F10, C1–C10), most of which were saxicolous (growing on granites), while only two were epiphytic (growing on *Quercus* sp.). The number of crustose (C1–C10) and foliose (F1–F10) lichens were equivalent, while no fruticose lichens were found at all (for detailed information see in Table 1). The differences in the number of lichen samples were mainly due to the limitation of the number of locations available for collection in the field. Lichen samples (one entire thalli of each individual) were collected using sterile blade and forceps and placed in sterile polythene bags, sealed and labelled. Samples were transported to the laboratory using an ice box and then stored at −80 °C for further processing. Lichen samples were washed with ultra-pure laboratory grade water to remove dirt and debris. To avoid any possible bias depending on the extreme sensitivity of lichens to environmental changes, all sampling procedures were conducted in one day and under similar microenvironmental conditions.

## DNA extraction, 16S rRNA gene amplification and sequencing

HiPure Soil DNA Kit was used to extract total genome DNA according to manufacturer's protocols (Magen Biotechnology Co., Ltd, Guangzhou, China). DNA concentration was monitored by Qubit® dsDNA HS Assay Kit. A total of 20–30 ng of DNA was used to generate amplicons containing V3–V4 hypervariable regions using a MetaVX Library Preparation kit (GENEWIZ, Inc., South Plainfield, NJ, USA). The forward primer contained the sequence 5′–7CCTACGGRRBGCASCAGKVRVGAAT–3′ and the reverse primer, 5′–GGACTACNVGGGTWTCTAATCC–3′, which were designed by GENEWIZ (Suzhou, China). The polymerase chain reaction (PCR) system contained 2.5 μl of TransStart buffer, 2 μl of dNTPs, 1 μl of each primer, 0.5 μl of TransStart Taq DNA polymerase, 20 ng template DNA, and ddH2O which was added to obtain the total volume of 25 μl. The PCR ran as follows: 3 min of denaturation at 94 °C, 24 cycles of 5s at 95 °C, 90 s of annealing at 57 °C, 10 s of elongation at 72 °C, and a final extension at 72 °C for 5 min. Indexed adapters were added to the ends of the amplicons by limited cycle PCR. Finally, the library was purified with magnetic beads. Paired-end (PE) sequencing was carried out with the Illumina Miseq Platform (Illumina, San Diego, USA) at Genewiz, Inc.

## Bioinformatic analysis

PRINSEQ (0.20.4) was used for sequence quality control. After quality filter (reads with length <200 bp) and removal of chimeric sequences, all sequences were grouped into operational taxonomic units (OTUs) using VSEARCH (1.9.6) (sequence similarity was set

**Table 1 Information of lichen samples.**

| Sample ID | Lichen species | Growth form | Substrate | Sampling site | Altitude (m) |
|---|---|---|---|---|---|
| F1 | *Phaeophyscia* sp. | Foliose | Granites | Scenic spot 116°7831′E, 40°5611′N | 283 |
| F2 | *Phaeophyscia* sp. | Foliose | Granites | Scenic spot 116°7832′E, 40°5612′N | 286 |
| F3 | *Canoparmelia* sp. | Foliose | Granites | Scenic spot 116°7832′E, 40°5611′N | 283 |
| F4 | *Candelaria asiatica* | Foliose | Granites | Nature reserve 116°7459′E, 40°5677′N | 753 |
| F5 | *Candelaria asiatica* | Foliose | Granites | Nature reserve 116°7392′E, 40°5669 | 860 |
| F6 | *Phaeophyscia* sp. | Foliose | Granites | Nature reserve 116°7392′E, 40°5668′N | 857 |
| F7 | *Phaeophyscia* sp. | Foliose | Granites | Nature reserve 116°7349′E, 40°5654′N | 912 |
| F8 | *Parmelia* sp. | Foliose | *Quercus* sp. | Nature reserve 116°7266′E, 40°5572′N | 1,115 |
| F9 | *Punctelia* sp. | Foliose | *Quercus* sp. | Nature reserve 116°7266′E, 40°5572′N | 1,115 |
| F10 | *Ramalina* sp. | Foliose | Granites | Nature reserve 116°7255′E, 40.5571′N | 1,130 |
| C1 | *Aspicilia cinerea* | Crustose | Granites | Nature reserve 116°763′E, 40°5652′N | 532 |
| C2 | *Aspicilia* sp. | Crustose | Granites | Nature reserve 116°7518′E, 40°5678′N | 705 |
| C3 | *Aspicilia cinerea* | Crustose | Granites | Nature reserve 116°7483′E, 40.5671′N | 721 |
| C4 | *Lepraria* sp. | Crustose | Granites | Nature reserve 116°7416′E, 40.5678′N | 813 |
| C5 | *Lecanora saxigena* | Crustose | Granites | Nature reserve 116°735′E, 40°5654′N | 912 |
| C6 | *Verrucaria funckii* | Crustose | Granites | Nature reserve 116°7348′E, 40°5649′N | 920 |
| C7 | *Bilimbia fuscoviridis* | Crustose | Granites | Nature reserve 116°7348′E, 40°565′N | 919 |
| C8 | *Buellia* sp. | Crustose | Granites | Nature reserve 116°7322′E, 40°5629′N | 958 |
| C9 | *Rhizoplaca* sp. | Crustose | Granites | Nature reserve 116°7255′E, 40°5571′N | 1,123 |
| C10 | *Aspicilia cinerea* | Crustose | Granites | Nature reserve 116°7255′E, 40°5571′N | 1,130 |

to 97%) against the SILVA v138 database. The OTU table was filtered to remove chloroplast sequences, and then OTUs were classified into different taxonomic levels using the Ribosomal Database Project (RDP) classifier (2.13). The names of annotated bacteria follow the List of Prokaryotic names with Standing in Nomenclature (LPSN) (*Parte et al.,*

2020). Alpha diversity indices (Chao 1 and Shannon), which indicate bacterial community richness and diversity, were calculated using Mothur software (1.35.1). Principal Co-ordinates Analysis (PCoA) and Unweighted Pair Group Method with Arithmetic Mean (UPGMA) based on the Bray-Curtis distance matrices were employed to determine beta diversity and cluster samples, respectively. Analysis of Similarities (ANOSIM) was applied to test the differences between bacterial communities associated with foliose and crustose lichens. Linear discriminant analysis Effect Size (LEfSe) was conducted using the Galaxy framework online tool (https://huttenhower.sph.harvard.edu/galaxy/), which can find microorganisms with significant differences between groups by calculating the contribution of each microbial abundance to the overall difference. Phylogenetic Investigation of Communities by Reconstruction of Unobserved States (PICRUSt2) was conducted to predict the function of lichen-associated microbiota based on the Kyoto Encyclopedia of Genes and Genomes (KEGG) and the Clusters of Orthologous Groups (COG) databases. All statistical analyses were performed with SPSS 22.0 (IBM Corp., Armonk, NY). Differences in phylum, order and genus relative abundances are presented as Means ± SE. Alpha diversity indices were calculated by using the Independent–sample t–test. A $P$-value < 0.05 was considered statistically significant, while a $P$-value < 0.01 indicated that differences were highly significant. The raw data obtained in this study have been submitted to the NCBI Sequence Read Archive under Bioproject number PRJNA972522.

## RESULTS

### 16S rRNA gene sequencing data statistics

After filtration of low-quality regions and removal of chimeras, a total of 1,232,513 effective sequences were obtained, with 36,494–79,179 for each sample and an average length of 442.04 bp (Table 2). Using cluster analysis, we identified 426 OTUs at 97% similarity level, including 16 phyla, 30 classes, 61 orders, 83 families and 125 genera.

The rarefaction curve (Fig. 1A) shows that the number of OTUs in the sample increases with sequencing depth, the curve finally flattens, indicating that the sequencing had covered most bacteria in the samples. Overall, the observed OTUs which stand for species richness in foliose lichens was higher than that of crustose lichens. The rank abundance curve (Fig. 1B) shows two aspects of diversity, namely richness and evenness. In general, the evenness and richness of the bacteria associated with foliose lichens was higher than that with crustose lichens.

### Composition of bacteria associated with foliose and crustose lichens

Under the conditions of this study, at the phylum level, the most dominant phylum of bacteria associated with foliose and crustose lichens was Pseudomonadota, with a relative abundance of 55.88 ± 4.45% and 69.76 ± 5.48%, in foliose and crustose, respectively (Fig. 2). The phyla Actinomycetota (foliose: 13.44 ± 2.61%; crustose: 9.46 ± 2.70%) and Acidobacteriota (foliose: 13.63 ± 1.83%; crustose: 12.15 ± 2.72%) were also present with a relatively high abundance. At the order level (Fig. 3), the dominant orders of bacteria were Acetobacterales and Hyphomicrobiales, with relative abundance of 4.44 ± 1.02%, 6.13 ±

**Table 2 Statistics of the sequencing data after filtering of each sample.**

| Sample | PE_reads | Effective tags | AvgLen (bp) | GC(%) |
|---|---|---|---|---|
| C1 | 105,344 | 76,922 | 440.06 | 52.15 |
| C2 | 86,141 | 60,796 | 441.5 | 53.45 |
| C3 | 95,084 | 68,337 | 439.64 | 53.01 |
| C4 | 94,928 | 67,785 | 445.24 | 53.33 |
| C5 | 93,315 | 66,617 | 442.93 | 51.93 |
| C6 | 100,889 | 75,061 | 441.09 | 50.72 |
| C7 | 106,163 | 79,179 | 441.79 | 52.59 |
| C8 | 79,509 | 49,814 | 444.23 | 55.31 |
| C9 | 103,014 | 69,208 | 441.5 | 53.09 |
| C10 | 81,332 | 50,249 | 439.71 | 52.46 |
| F1 | 82,675 | 45,557 | 440.62 | 52.52 |
| F2 | 78,370 | 39,679 | 444.18 | 54.12 |
| F3 | 73,996 | 36,494 | 439.39 | 53.41 |
| F4 | 82,112 | 58,093 | 441.96 | 53.71 |
| F5 | 90,903 | 62,486 | 443.08 | 53.62 |
| F6 | 98,442 | 68,661 | 443.19 | 53.76 |
| F7 | 86,286 | 58,717 | 445.69 | 54.71 |
| F8 | 81,252 | 55,099 | 444.29 | 54.65 |
| F9 | 101,652 | 70,326 | 439.45 | 52.72 |
| F10 | 113,565 | 73,433 | 441.27 | 52.10 |

Note:
PE (Paired-End) reads: number of original PE reads; Effective tags: number of valid sequences after chimera removal over the original number of PE reads; AveLen (bp): average length of valid sequences; GC (%): GC content of valid data.

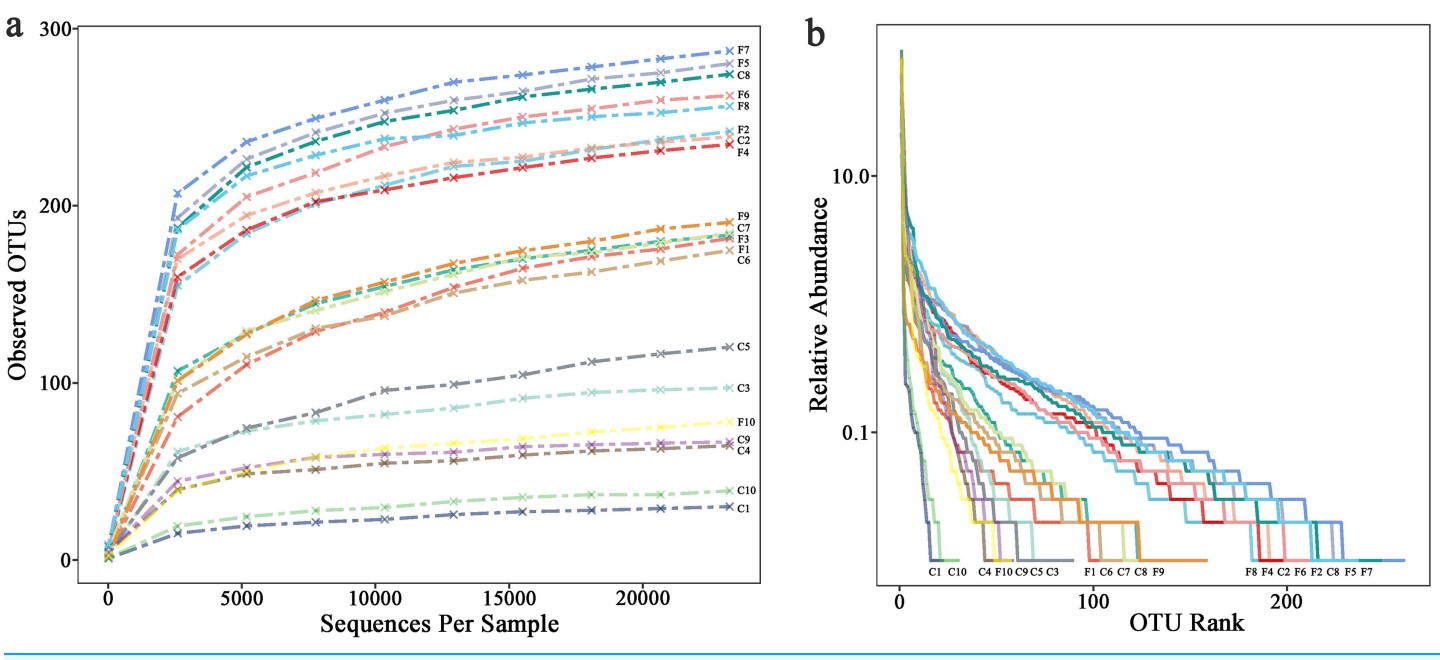

**Figure 1** (A) Rarefaction curves and (B) rank abundance curve.
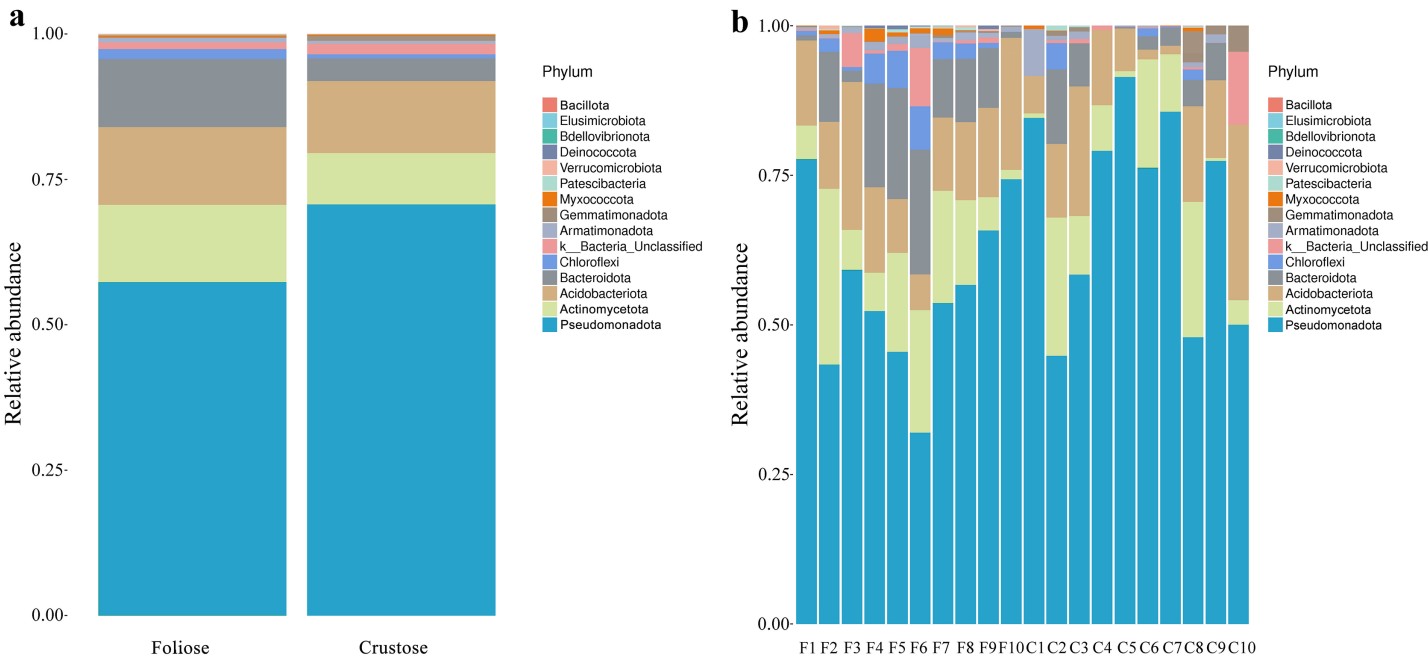

**Figure 2 Distribution of bacteria at the Phylum level associated with foliose and crustose lichens.** (A) For groups, (B) for each sample.

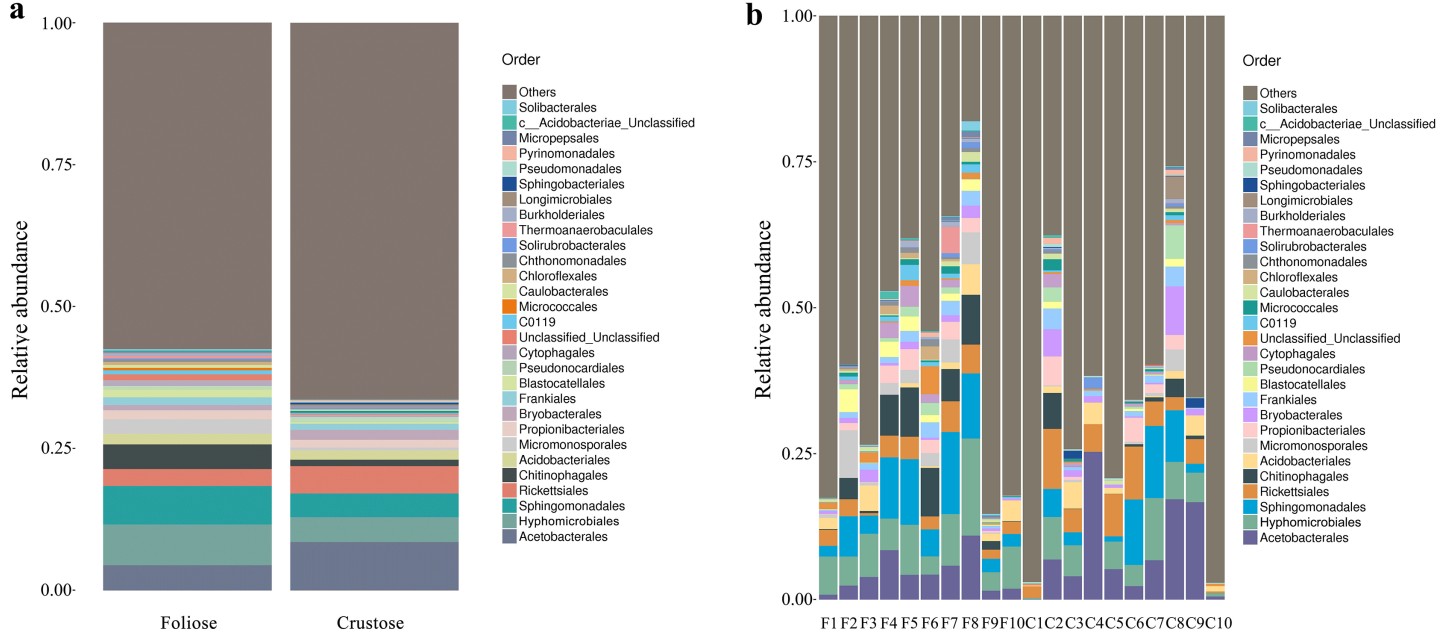

**Figure 3 Distribution of bacteria at the Order level associated with foliose and crustose lichens.** (A) For groups, (B) for each sample.

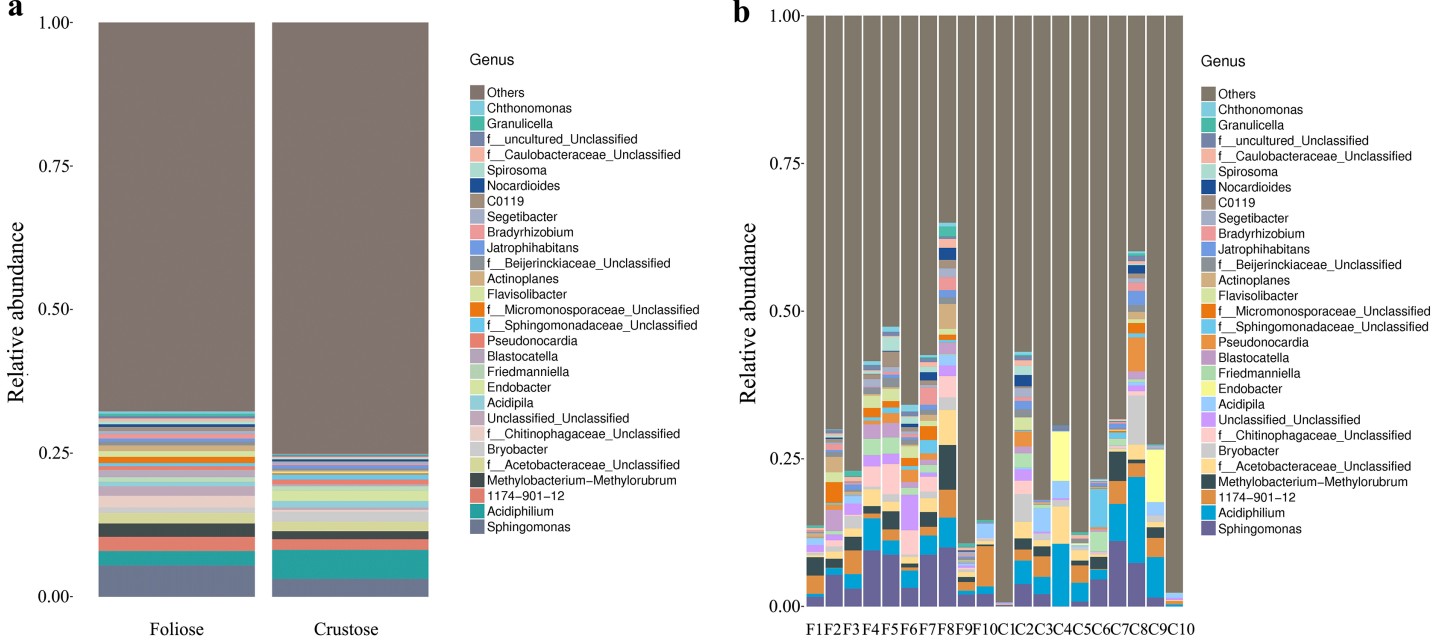

**Figure 4 Distribution of bacteria at the Genus level associated with foliose and crustose lichens.** (A) For groups, (B) for each sample.

0.71% for foliose lichens and 6.62 ± 2.11%, 4.39 ± 1.08% for crustose lichens. At the genus level (Fig. 4), *Sphingomonas* (foliose: 5.44 ± 1.09%; crustose: 3.14 ± 1.16%) and *Acidiphilium* (foliose: 2.5 ± 0.54%; crustose:3.99 ± 1.14%) had higher relative abundance in both groups of lichens.

## Diversity of bacterial communities associated with foliose and crustose lichens

Under the conditions of this study, Alpha diversity results showed that the Chao1 indices of foliose and crustose lichens (Fig. 5a) were 246.57 ± 20.42 and 151.40 ± 30.38, respectively ($P = 0.02$). The Shannon indices of bacterial communities associated with foliose and crustose lichens (Fig. 5b) were 3.69 ± 0.55 and 2.71 ± 0.56 respectively but did not show significant differences between the two groups ($P = 0.43$).

The UPGMA tree (Fig. 6) and the PCoA (Fig. 7) unraveled the microbial composition between different samples. Performing ANOSIM analysis, we found a significant difference between the community structures of bacteria associated with foliose and crustose lichens ($P = 0.02$).

## Taxon differences of lichens-associated bacteria between foliose and crustose lichens

According to phylogenetic map (Fig. 8), there are nine bacterial taxa showed significant differences between foliose and crustose lichens in this study. Among the bacteria associated with foliose lichens, the relative abundance of Chloroflexi, Bacteroidia, Bacteroidota, Micromonosporae, Micromonosporales, Chitinophagae, Chitinophagales, Blastocellaceae, Blastocellales was significantly higher than that of bacteria associated with

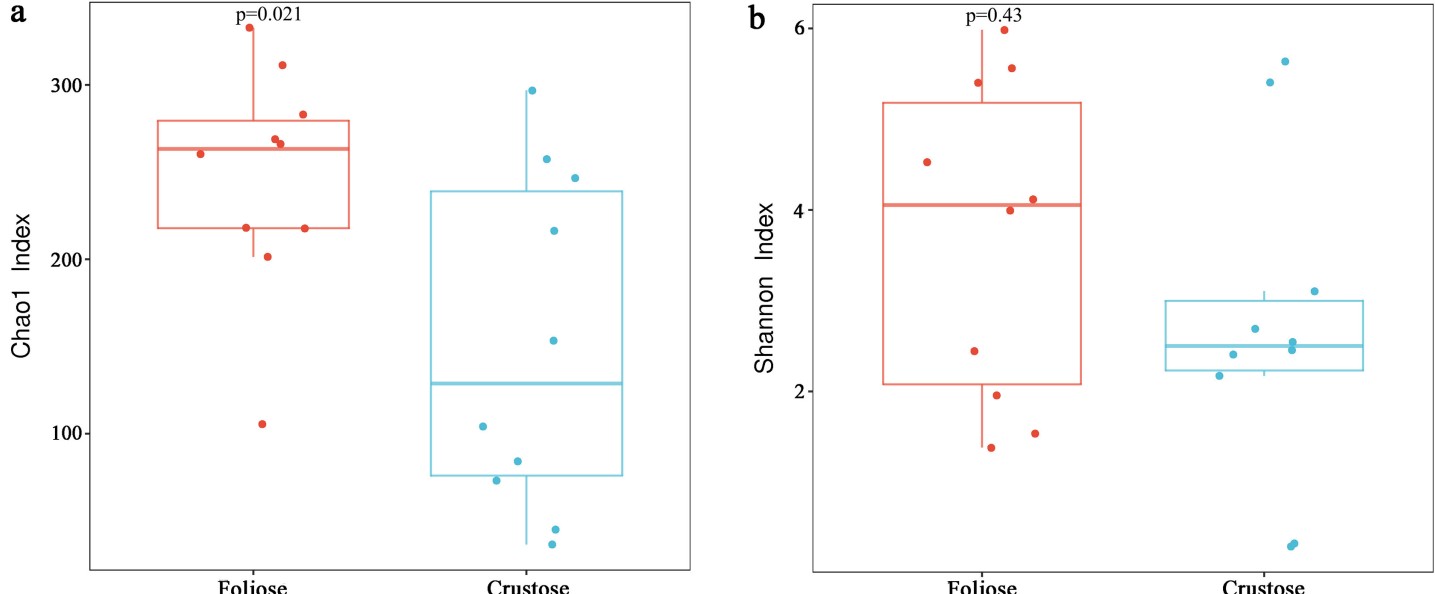

**Figure 5 Boxplot of (A) Chao1 and (B) Shannon diversity indices referred to the bacterial communities associated with foliose and crustose lichens.** Boxes represent the interquartile range (IQR; between 25th and 75th percentiles), horizontal line inside the box defines the median, outliers greater than 1.5 and less than three times the IQR.

crustose lichens. No taxon was found to have a significantly higher relative abundance with crustose lichens than foliose lichens.

## PICRUSt2 gene function predictions

Functional annotations were based on KEGG and COG databases. Most of the bacterial gene functions were related to metabolism (Fig. 9A), such as the metabolisms of carbohydrates, amino acids, vitamins, and cofactors. The abundance of genes related to amino acid transport and metabolism, function unknown, *etc.* were high (Fig. 9B). No significant difference in the functional abundance between associated bacteria of foliose and crustose lichens was found under the conditions of this study.

## DISCUSSION

The diversity and composition of bacterial communities associated with foliose and crustose lichens in Mt. Yunmeng were examined using 16S amplification sequencing. Our results showed that the dominant phyla of bacteria associated with these two types of lichens were Pseudomonadota, Acidobacter, Actinomycetota, which were similar to those described by other studies on lichens (*Swamy & Gayathri, 2021*; *Grube et al., 2015*; *Bjelland et al., 2011*). Acetobacterales and Hyphomicrobiales were the most abundant orders of lichen-associated bacteria, which were also reported by other studies (*Bates et al., 2011*; *Hodkinson & Lutzoni, 2009*).

Our results suggest that the growth form of lichens has an impact on the bacterial communities associated with lichens, which is similar to results published by *Park et al. (2016)*. Firstly, alpha diversity indices showed that the richness of associated bacteria in foliose lichens was significantly higher than that in crustose lichens (Fig. 5), which may be

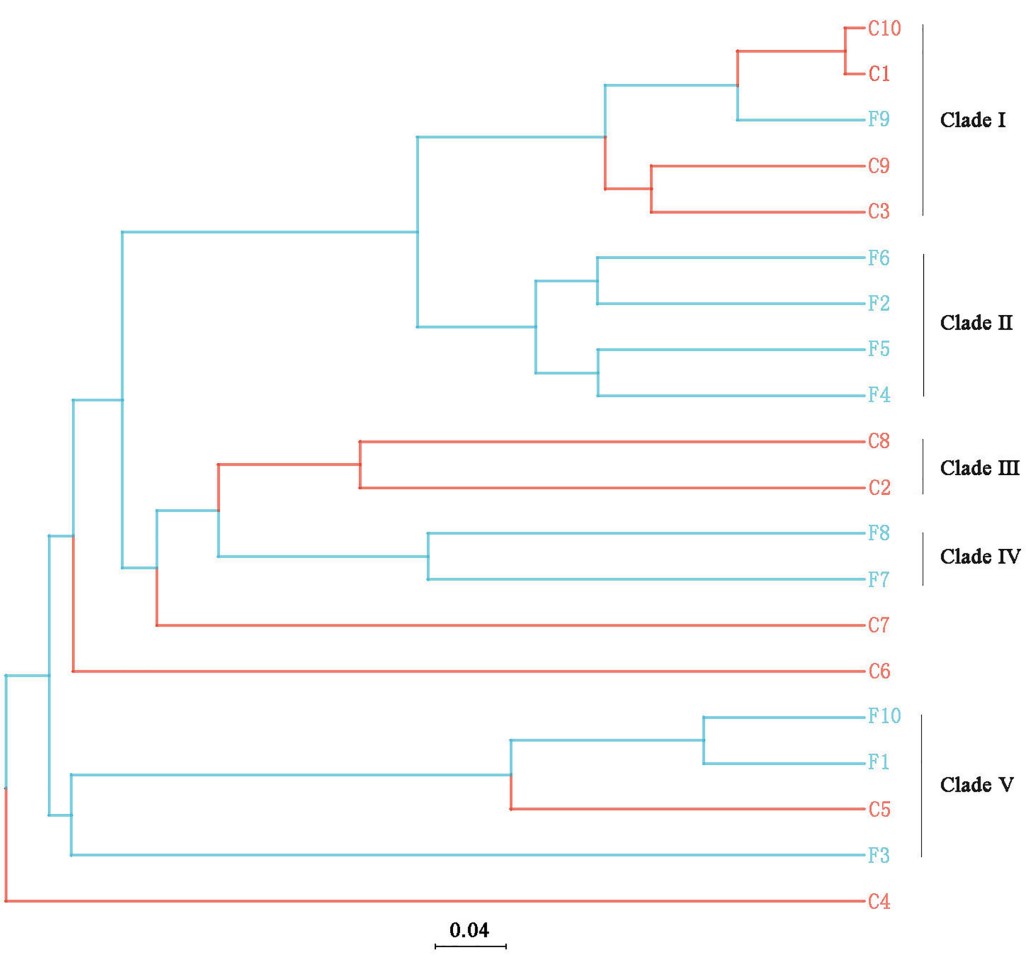

**Figure 6 UPGMA cluster analysis based on Bray-Curtis distances with five clades being recognized.** The bar below the tree represents the scale of distances, a closer distance and shorter branch length represents higher similarity of the species composition between two samples.

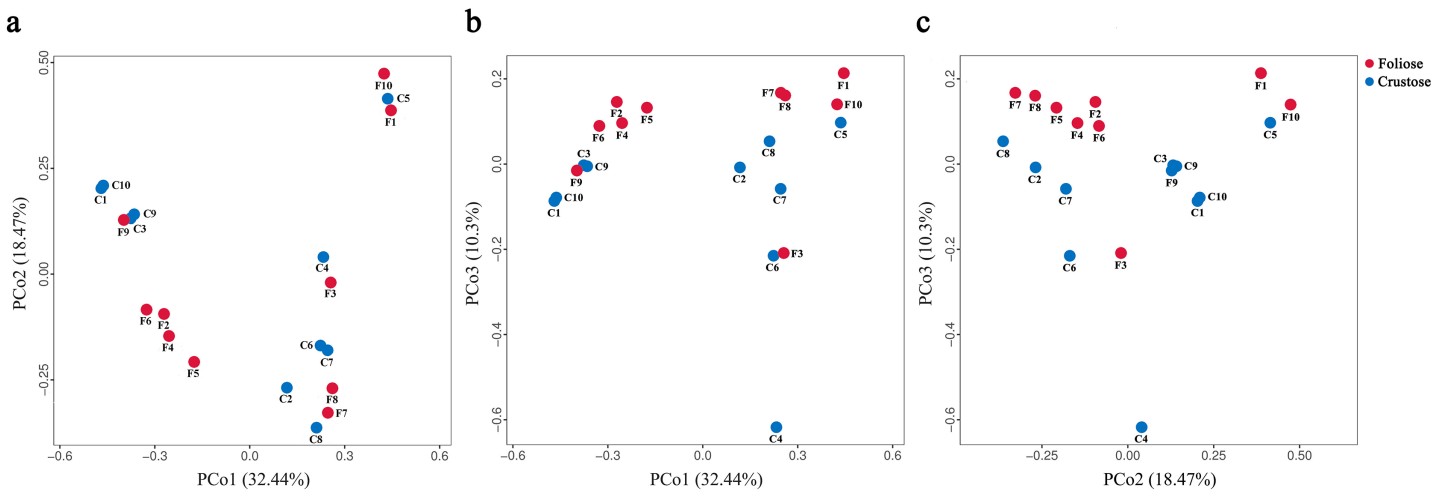

**Figure 7 PCoA results for (A) PCo1 *vs* PCo2, (B) PCo1 *vs* PCo3, and (C) PCo2 *vs* PCo3.** Percentages of total variation explained by each axis are shown in brackets.

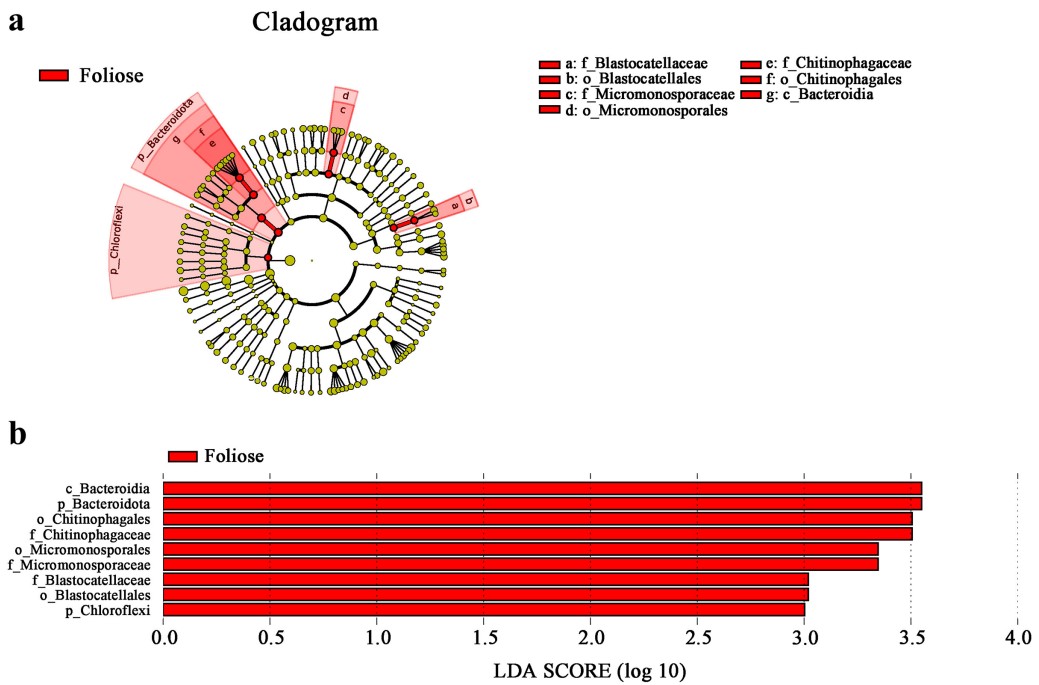

**Figure 8 LefSe analysis.** (A) Cladogram, and (B) LDA value distribution histogram. From the inside to the outside: phylum, class, order, family, and genus level. Different color points in the phylogenetic tree represent bacteria which are significantly different between the two groups of lichens. Yellow points indicate bacteria that show no significant difference between the two groups. LDA score was set to be at 3.0.                                                 

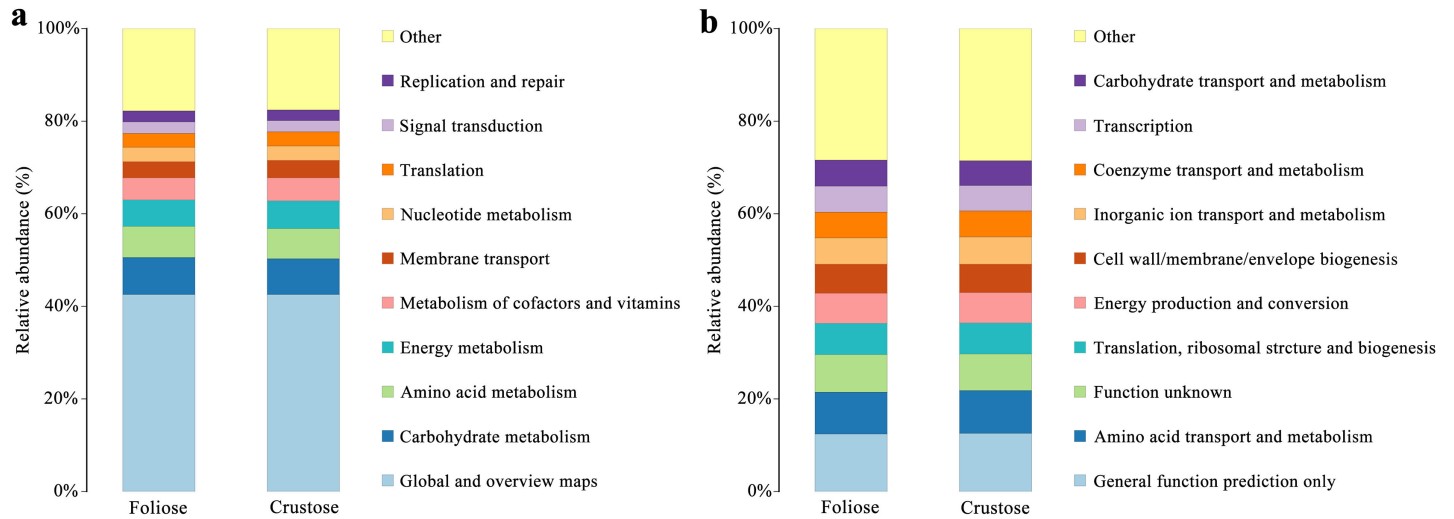

**Figure 9** Functional annotation for non-redundant gene sets based on the functional classification of (A) KEGG (level 2) and (B) COG database.                                 

attributed to the more complex thalli structure of foliose lichens compared to that of crustose lichens (*Nash, 2008*). The fact that the powdery and the most poorly structured crustose lichen *Lepraria* sp. (C4) clustered at the most outer branch of the UPGMA tree (Fig. 6) supports this suggestion. Secondly, in the cluster analysis foliose and crustose

lichens were grouped into several different clades (Fig. 6). On the UPGMA tree, five distinct clades were recognized, with Clade I and III mainly comprising crustose lichens, while Clade II, IV and V comprising foliose lichens. Thirdly, PCoA results showed a significant difference between beta diversity of foliose lichen-associated bacterial communities and that of crustose lichen-associated communities (Fig. 7). Combining these findings suggest that the growth form of lichens had a strong influence on the bacterial community composition.

Moreover, it seems that the lichen species itself may affect the bacterial community it is hosting. Among the materials we collected, there were three species with at least two samples (Table 1), *i.e.*, *Phaeophyscia* sp. (F1, F2, F6, F7), *Aspicilia cinerea* (C1, C3, C10) and *Candelaria asiatica* (F4, F5). Two samples of each species clustered together in the UPGMA tree, namely F2 and F6 for *Phaeophyscia* sp., C1 and C10 for *A. cinerea*, and F4 and F5 for *C. asiatica* (Fig. 4). Since the composition of microbial communities shows some species specificity under the conditions of this study, we speculate that it is likely to be affected by the host lichen. However, not all samples of the same lichen species clustered together. For example, though belonging to the same species *Phaeophyscia* sp., F1 and F7 were located at branches far from the clade formed by F2 and F6 (Fig. 4), suggesting that the species specificity of bacterial communities is not robust. Most of the lichens we collected were saxicolous, and only two samples were epiphytic (F8 and F9), which were scattered among the others in the UPGMA tree (Fig. 6). Based on the existing data, we therefore speculate that the substrate may have no or little impact on the lichen-associated bacteria, a result however, that requires further research.

Bacteria in lichen-hosted microbial communities were considered to have ecological and physiological roles and are indispensable for the lichen thallus (*Grube et al., 2009*). In this study, most of the gene functions annotated by the lichen-associated bacteria were related to metabolism (Fig. 9), suggesting that the related bacteria may help to improve the nutritional balance of their hosts. The associated bacteria orders, Hyphomicrobiales and Acetobacterales are well known to be able to fix nitrogen in microbe-plant interactions (*Garrity, Bell & Lilburn, 2005*; *Saravanan et al., 2008*). They were also detected in samples investigated by other studies and were recognized as an important symbiotic component that may provide fixed nitrogen to lichen ecosystems (*Bates et al., 2011*; *Hodkinson & Lutzoni, 2009*). Besides, these two orders of bacteria are also known to perform other crucial functions supporting the symbiosis, such as providing auxin and vitamins, and thus helping to protect the lichens from physiological stress (*Erlacher et al., 2015*; *Cernava et al., 2017*). *González et al. (2005)* studied the culturable Actinomycetes isolated from lichens and found half of the isolates could produce antibacterial substances. Moreover, a strain of Actinomycete isolated from *Cladonia uncialis* was found to be able to produce a new compound with strong antibacterial activity (*Davies et al., 2005*). In addition, *Bhatti, Haq & Bhat (2017)* showed that Actinomycetes play a role in eliminating harmful microorganisms in soil, suggesting that they might play a defensive role in lichens. Therefore, most of these lichen-associated bacteria play an irreplaceable and important role in the colonization and growth of lichens.

## CONCLUSIONS

The relationship between the microbes and their lichen hosts has intrigued deliberate rethinking of the concept of lichen symbiosis (*Allen & Lendemer, 2022*). It seems that microbes are being part of the lichen itself, we should notice that some bacteria were persistently found to co-exist with lichens in a series of studies and might therefore play an essential role in the microbial community. Therefore, it is reasonable to view some lichen-associated bacteria—if not all—as integrated and inseparable members of a symbiotic ecosystem that plays a significant role in sustaining this mutualism. In this study, in spite of its exploratory approach, we found quite convincing evidences that the growth form of lichens has a strong effect on the composition of their microbial communities. In reverse conclusion, we propose another perspective, *i.e.*, considering the possibility that microbial communities may also play a key role in determining the growth form of their hosts. Since we have not investigated this particular question in our study, we strongly recommend this possibility to be addressed in future research.

### Funding

This work was financially supported by the Beijing Government, and Mengya Plan (Grant No. 23CE-BGS-06) from the Beijing Academy of Science and Technology, except for the sampling expenditure, which came from the personal finances of Manrong Huang.
The funders had no role in study design, data collection and analysis, decision to publish, or preparation of the manuscript.

### Grant Disclosures

The following grant information was disclosed by the authors:
Beijing Academy of Science and Technology: 23CE-BGS-06.
Personal finances of Manrong Huang.

### Competing Interests

The authors declare that they have no competing interests.

### Author Contributions

- Yimeng Li performed the experiments, analyzed the data, prepared figures and/or tables, authored or reviewed drafts of the article, and approved the final draft.
- Yinzhi Huang performed the experiments, analyzed the data, prepared figures and/or tables, and approved the final draft.
- Torsten Wronski analyzed the data, authored or reviewed drafts of the article, and approved the final draft.
- Manrong Huang conceived and designed the experiments, authored or reviewed drafts of the article, and approved the final draft.

## Data Availability

The raw data obtained in this study have been submitted to the NCBI Sequence Read Archive under Bioproject number PRJNA972522.

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
