# Peer review of "Diversity of bacteria associated with lichens in Mt. Yunmeng in Beijing, China"

_PeerJ, doi:10.7717/peerj.16442_

## Round 0.1 · original submission · Major Revisions

Dear authors, please revised the manuscript as suggested by the reviewers. When submitting your revised manuscript, please also submit a point-by-point response letter.

Reviewer 1 ·

Basic reporting

I believe this article would improve if it were revised by someone who speaks English as a first language. There are subject-verb disagreements throughout, and some sentences are misleading if interpreted exactly as written.

The first paragraph of the results section, to me, reads like it should be part of the methods section.

Experimental design

The difference in sampling intensity between the scenic spot and the nature reserve, in addition to the sampling intensity between epiphytic lichen and saxicolous lichen might be worth more explanation. Was the difference in sampling due to availability of samples to collect, or was it limited by time to collect samples?

In the last paragraph of the introduction, the objectives are stated, which mention mapping lichen diversity in Beijing. I don't immediately see the connection between bacterial communities on foliose vs crustose lichen with these objectives- I think the objectives should be adjusted to more closely match the experimental design.

Briefly in the introduction, and in the discussion, there is text that mentions the potential impact of tourism on microbial communities in lichen. It seems that the imbalance in sample collection (3 vs 17) does not make for a strong experimental design to ask or answer this question.

I might have missed it, but Figure 3 shows a p value, presumably testing the difference between chao 1 and shannon index between crustose and foliose growth forms. What statistical test was used?

Validity of the findings

I would urge caution for making broad statements based on this study. However, I do want to applaud the researchers for doing this interesting research that serves as an introductory look at bacterial communities on two lichen growth forms. Very interesting work.

For Figure 5, I think it might be easier to interpret this if only PC1 and PC2 axes are shown. While the authors are editing this figure, could you indicate which samples were collected in the scenic area vs nature reserve? Also, could the 2 epiphytic samples be indicated apart from the saxicolous lichen? I think this would help the reader quickly view the sample to sample variation in community composition, as well as the dissimilarity of the bacterial communities between the two growth forms.

I believe the findings are valid and that the sequencing was performed correctly. However, as stated currently, I don't think the results of this work support the interpretations that are stated in the discussion. Please see highlighted pdf document which points to confusion on whether saxicolous and epiphytic lichen differ in their bacterial community composition.

Additional comments

The sample size (n= 10 for crustose, n=10 for foliose) presents a question of how figure 2 was generated. Did the authors take the average abundance across the 10 samples for a) Phylum, b) Order, and c) Genus? Instead of showing aggregated values, I think it would be more interesting to see the composition of bacterial phyla, order, and genera for each individual sample. This would allow readers to view the variation across the 10 samples of each growth form, which I see as the strongest part of this research.

In Figure 3, It is nice to see individual points for each sample- Is it worth interpreting why the chao 1 index has a higher spread for crustose, whereas the shannon index has a wider spread for Foliose?

Annotated reviews are not available for download in order to protect the identity of reviewers who chose to remain anonymous.

Reviewer 2 ·

Basic reporting

The manuscript's subject matter is highly intriguing, and its data would make a valuable contribution to the existing literature on bacteria associated with lichens. However, due to flaws in the research design, the results and discussion are of low quality. I recommend that authors reanalyze the data and focus on producing a concise and polished paper.

Experimental design

Some major flaws that should be addressed.
First, a rigorous categorization of lichen growth forms is need. For example, all Ramalina species are fruticose lichens, and most Rhizoplaca species are foliose (squamulose) rather than crustose lichens. The data on them must be deleted.
Second, lichens are extremely sensitive to environmental (and microclimates) changes. As poikilohydric organisms, lichens' (including associated bacteria) physiological processes, such as carbon gain, are dependent on hydration. To ensure the reliability of results, lichen sampling should be conducted over a short time and under similar microenvironment conditions (e.g. similar weather). In this manuscript, lichens were collected within a small region, the authors could control these conditions and finish the work in several days.
Third, the experiment procedures should be described in enough detail to allow others to replicate them. For example, sterile tool? distance range between samples? One or some thallus removed from substrate? Single or mixed sample? sample storage and transported to the laboratory (at −20 °C)? debris removal? and so on. Rock type and host tree species should also be stated.
Finally, there were too few repeats between different treatments to subject them to statistical analysis.

Validity of the findings

1. I was very surprised when I read that Cyanobacteria were dominant phyla (RA>0.5!!!) in these lichens with green algae. I doubt this result is correct here. In my experience, this is possibly because the OTU table is not filtered to remove chloroplast sequences. Cyanobacteria are considered the ancestors of chloroplasts, and chloroplasts is as cyanobacteria at the phyla level. Re-examination and recalculation of your data would be more appreciate. All relevant results and discussions must be reorganized.
2. Again. Too few repeats between different treatments. Relevant results failed to confirm the influence of studied factors on bacteria associated with lichens, such as lichen species, altitude, substrate type and tourists. Consider rewriting the results, discussion, and conclusion section. ONLY focus on bacterial communities of two lichen growth forms.

Additional comments

L18 & L145: Acidobacter? Or Acidobacteria? Different spelling in Figure 2a.
L25-27: The results in gene function prediction were not shown in Abstract, which resulted in a lack of logical coherence between the preceding and following sentences.
L93-95: Primer name?
L105-120: A detailed and clear description of analysis methods is necessary. ALSO, the software version number needs to be specified clearly.
L114: “ANOSIM” would be better.
L143 and so on: meaning of the numbers (e.g., 64.22% ± 6.74%...)? mean ± SD or mean ± SE? ‘’64.22 ± 6.74% ‘’would be better for shown.
L151: Acidophilia? Or Acidiphilium? See Figure 2c.
L155: Figure 3: based on ANOVA or non-parametric test? Stated in Analysis section!
L155-165: Seem redundant. The differences between the two lichen groups were easy to understand, but some sentences didn't make sense.
L161-163: That sentence doesn't make sense. The results must be clear.
L168-169: Move to Analysis section.
L181: KEGG and COG, full name must be provided upon initial mention.
Discussion SECTION: should be based on the data recalculation.
L197-221: Poor discussion without any reference.
L261-264: So ambiguous inference.
L305: “meta-omics”

Reviewer 3 ·

Basic reporting

BASIC REPORTING

Clear, unambiguous, professional English language used throughout.

- Although I am not a native English speaker, it seems to me that the English, and the writing style in general, should be improved in several places. There are several sentences in the manuscript that appear defective in logical, syntactic and/or grammatical structure, which without careful revision are difficult to interpret or in some cases meaningless. Some of these sentences are noted in the attached pdf file, but in general a careful revision by a native English speaker would be desirable.


Intro & background to show context.
- Ok.


Literature well referenced & relevant.
- Ok.


Structure conforms to PeerJ standards, discipline norm, or improved for clarity.
- No comment.


Figures are relevant, high quality, well labelled & described.

- The graphic appearance of the figures could be much improved, making all captions more concise and readable. Further comments can be found in the attached pdf.


Raw data supplied (see PeerJ policy).
- Yes.

Experimental design

EXPERIMENTAL DESIGN

Original primary research within Scope of the journal.
- Yes.


Research question well defined, relevant & meaningful. It is stated how the research fills an identified knowledge gap.

- Although quite interesting data and insights are found in this manuscript, the purpose of the research in my opinion is not sufficiently made explicit, and consequently also the sampling design appears in many respects vague and inadequate to support some of the most relevant conclusions proposed in this work, particularly those referring to the ecological aspects of the research.
At the end of the introduction (lines 56-59), where one would normally expect to find a clear list of objectives on which the research was developed, it is written that: "While we were trying to map lichens biodiversity in Beijing, it occurred to us that we could also evaluate such factors using our materials from Mt. Yunmeng, and therefore enhance our understanding of the maintenance and dynamics of the microbial communities and the relationship with their hosts."
It thus appears that the research was conducted with a purely exploratory purpose, which is reflected in a certain randomness of the field sampling procedures. For example, in lines 81-86, it is stated that out of 20 surveys only 3 were conducted in the most heavily anthropised areas, a number which in my opinion is not sufficient to support subsequent discussions on the role played by anthropisation on symbiotic bacterial communities. The same applies to the analysis referring to the role of the substrate, based on a comparison between 18 saxicolous species and only 2 epiphytes (lines 225-228), a numerical imbalance that in my opinion does not allow any kind of statistical analysis to be validated for this comparison. The analysis referring to the altitudinal gradient, which is partly based on the comparison of a few samples that, although not identified at species level, were treated as belonging to the same species (lines 213-214 and 219), also does not seem to be robustly supported by this experimental protocol.


Rigorous investigation performed to a high technical & ethical standard - Methods described with sufficient detail & information to replicate.

- Ok as for the general methods of analysis, less so the sampling criteria, as already reported.

Validity of the findings

Impact and novelty not assessed.
- No comment.


Meaningful replication encouraged where rationale & benefit to literature is clearly stated.
- No comment.


All underlying data have been provided; they are robust, statistically sound, & controlled.
- Ok, except for weaknesses indicated in the other sections.


Conclusions are well stated, linked to original research question & limited to supporting results.

- I really appreciate the authors' contribution regarding the composition of the bacterial communities analysed and the possible role played by growth forms (e.g. comparison between foliose and crustose species). However, other conclusions of this work (e.g. effects of altitude, substrate and anthropic disturbance on the composition of symbiotic bacterial communities), as argued above, would need a clearer sampling design, specifically aimed at testing the proposed hypotheses. In the absence of such a design, these conclusions are unsupported by the investigation conducted, resulting in a significant decrease in the value of the proposed research.
I therefore suggest defining the research objectives more clearly, structuring a sampling design consistent with these objectives, supplementing the collected data (if necessary) with a new sampling campaign, and finally revising the collected data, verifying or disproving the hypotheses on which the authors decide to structure the work.
If such revision and integration work is carried out, I believe that the authors will be able to produce a very interesting contribution to shed new light on the symbiotic bacterial communities of lichen ecosystems and their relationship with the various environmental and anthropic factors mentioned.

Additional comments

Please find attached a pdf file with detailed comments on manuscript and figures.

Annotated reviews are not available for download in order to protect the identity of reviewers who chose to remain anonymous.

---

## Round 0.2 · Major Revisions

Please revise the manuscript carefully and thoroughly by following the reviewers' comments. When re-submitting your manuscript, please provide a point-by-point response letter to address all the comments thoroughly with a track-change manuscript. Thanks.

Reviewer 1 ·

Basic reporting

I do not feel as though the authors were genuine in their revisions of this paper.

I still see many English language issues. I also see that some of my comments from the pdf were not carried over, i.e. the authors still choose to use the term 'self-sufficient' to describe an ecosystem that lichen are a part of.

Furthermore, I am puzzled at the response to my comment that 3 vs 17 does not make for a strong experimental design, to which the authors responded that they simply inferred a conclusion based on the 3 vs 17 comparison. This doesn't seem scientific.

"Comment: Briefly in the introduction, and in the discussion, there is text that mentions the potential impact of tourism on microbial communities in lichen. It seems that the imbalance in sample collection (3 vs 17) does not make for a strong experimental design to ask or answer this question.
Answer: Based on the reality of our sampling, only 3 valuable lichen samples from scenic spots were obtained, so based on the results of the current study, we infer this conclusion"

This is dangerous because conclusions about the 3 samples from the scenic spot are being compared to the 17 samples from the nature reserve- and being used to say that human activities could influence bacterial composition on the lichen. At a 3 vs 17 comparison, difference could be attributed to many things which is why I don't think it is constructive to infer anything from the scenic vs non-scenic area.

Experimental design

The authors made several improvements to the methods section.

Validity of the findings

The genetic sequences that were observed are valid, but I hesitate to make any conclusions about what bacterial communities can be found on lichen elsewhere based on this study. This is because there is a large number of factors that impact bacterial diversity, but it seems like the authors are suggesting that the bacterial diversity observed in these 20 samples is valid to represent a worldwide understanding of lichen bacterial communities. Perhaps using the language 'under the conditions of thus study' would make me feel more comfortable with what the findings van speak to with validity.

Additional comments

This work could be edited by person with a scientific background who speaks English as a native language. I still noticed subject verb disagreements, anthropomorphism (i.e. self-sufficient ecosystem), and language that is too conversational, particularly in the conclusion "But why should we not take another perspective" line 272.

Annotated reviews are not available for download in order to protect the identity of reviewers who chose to remain anonymous.

Reviewer 3 ·

Basic reporting

The authors made most of the requested changes and the manuscript is considerably improved. There are, however, in my opinion, still some points that need to be improved. In particular, since many of the doubts raised by the reviewers might be shared by some readers, it is important that some of the clarifications that were given by the authors in their rebuttal document are fully incorporated into the manuscript.
Further details below.

Lines 58-64: The authors were recommended to state more clearly the objectives of the work. Although the sentence at the end of the introduction has been partly reworded, it seems to me that there is still no explicit listing (e.g. by bullets) of the research objectives. I therefore suggest amending and supplementing the paragraph at lines 58-64 as follows (or in a similar manner):
At line 58: After "... Villareal, 2022)." Full stop, new paragraph:
"When we were mapping the lichens biodiversity of Beijing Municipality, a heavily urbanized ecosystem with a degraded lichens diversity, we also collected materials from Mt. Yunmeng, an area that was gradually restored in recent years (http://www.yunmengshan.org.cn/ymsjqjs). Here, we glimpsed the rare opportunity to evaluate the diversity of lichen-associated bacterial communities in an ecosystem during its restoration process, and therefore enhance our understanding of the maintenance and dynamics of lichen-associated microbial communities and the relationship with their hosts. To this aim, we conducted DNA extraction, 16S rRNA gene amplification and sequencing on the material gathered, with the following objectives: i. To compare the composition of the associated microbial communities between two growth form lichens (saxicolous vs epiphytic); ii. To evaluate to what extent the anthropic impact may affect these bacterial communities, by comparing records from a scenic spots vs those from a nature reserve.

Line 86: In the sentence "Lichen samples were identified refer to Brodo et al (2001) and Nash et al (2007)." check if it sounds better replacing the word "refer to" with "referring to", or with "with reference to".

Line 91: A clarification on the imbalance in the number of surveys and types surveyed should be included here, considering the answers given by the authors in their rebuttal document. For example, after "... see in Table 1).", the authors could specify "The differences in the number of lichen samples were mainly due to the limitation of the number of locations available for collection in the field.", as explained in the rebuttal.

Line 94: After "...debris.", as written in the answer to reviewer 2, the authors could specify that all the sampling procedure was conducted in one day and under similar microenvironment conditions, in order to avoid any possible bias depending on the extreme sensitivity of lichens to environmental changes.

Lines 113-116: Here it might be appropriate to state explicitly, as indicated in response to reviewer 2, that the OTU table was filtered to remove chloroplast sequences, thus avoiding any possible error in assessing the contribution of Cyanobacteria.

Line 165: Consider the opportunity to use the first capital letter in reference to the Chao1 index.

Line 212: change "may affects" to "may affect".

Lines 214/216/220. The authors should better specify the reason why samples F1, F2, F6 and F7, although not identified at species level, can still be reasonably attributed to the same species.

Lines 270-272: In their rebuttal document, the authors have specified in several points that this study was a first exploratory research, which could partly justify some concerns proposed by reviewers. This limit, however, should be also declared to the reader. For instance, the sentence "In this study, we found that the growth form of lichens has a great effect on the composition of their microbial communities.", could be changed as "In this study, in spite of its exploratory approach, we found quite convincing evidences that the growth form of lichens has a great effect on the composition of their microbial communities." - or similar.

Caption to Fig. 5: " Boxplot of (a) Chao1 and (b) Shannon diversity indices refer to" - Change "refer to" with "referred to".

Fig. 5: In both Figs. a and b, please move the value of p as not to cover datapoints.

Fig. 6: Please give a better proportion to the bar (0.04) and specify in the caption if it represents the scale of distances.

Fig. 7 (and caption): in order to avoid any confusion with PCA, the axes should be given not as PC1-PC2-PC3, but as PCo1, PCo2, PCo3 (or simply as Axes 1, 2, 3).

Experimental design

-

Validity of the findings

-

Additional comments

-

---

## Round 0.3 · accepted · Accept

After going through the rebuttal letter and the revisions of the manuscript, I think that the manuscript is acceptable.